# The Multicellular Effects of VDAC1 N-Terminal-Derived Peptide

**DOI:** 10.3390/biom12101387

**Published:** 2022-09-28

**Authors:** Uttpal Anand, Anna Shteinfer-Kuzmine, Gal Sela, Manikandan Santhanam, Benjamin Gottschalk, Rajaa Boujemaa-Paterski, Ohad Medalia, Wolfgang F. Graier, Varda Shoshan-Barmatz

**Affiliations:** 1Department of Life Sciences, Ben-Gurion University of the Negev, Beer-Sheva 84105, Israel; 2National Institute for Biotechnology in the Negev, Ben-Gurion University of the Negev, Beer-Sheva 84105, Israel; 3Gottfried Schatz Research Center, Division of Molecular Biology and Biochemistry, Medical University of Graz, 8010 Graz, Austria; 4Department of Biochemistry, University of Zurich, 8057 Zurich, Switzerland; 5BioTechMed Graz, 8010 Graz, Austria

**Keywords:** apoptosis, mitochondria, peptide array, protein–protein interaction, voltage-dependent anion channel-1

## Abstract

The mitochondrial voltage-dependent anion channel-1 (VDAC1) protein functions in a variety of mitochondria-linked physiological and pathological processes, including metabolism and cell signaling, as well as in mitochondria-mediated apoptosis. VDAC1 interacts with about 150 proteins to regulate the integration of mitochondrial functions with other cellular activities. Recently, we developed VDAC1-based peptides that have multiple effects on cancer cells and tumors including apoptosis induction. Here, we designed several cell-penetrating VDAC1 N-terminal-derived peptides with the goal of identifying the shortest peptide with improved cellular stability and activity. We identified the D-Δ(1-18)N-Ter-Antp comprising the VDAC1 N-terminal region (19–26 amino acids) fused to the Antp, a cell-penetrating peptide. We demonstrated that this peptide induced apoptosis, autophagy, senescence, cell volume enlargement, and the refusion of divided daughter cells into a single cell, it was responsible for reorganization of actin and tubulin filaments, and increased cell adhesion. In addition, the peptide induced alterations in the expression of proteins associated with cell metabolism, signaling, and division, such as enhancing the expression of nuclear factor kappa B and decreasing the expression of the nuclear factor of kappa light polypeptide gene enhancer in B-cells inhibitor, alpha. These cellular effects may result from the peptide interfering with VDAC1 interaction with its interacting proteins, thereby blocking multiple mitochondrial/VDAC1 pathways associated with cell functions. The results of this study further support the role of VDAC1 as a mitochondrial gatekeeper protein in controlling a variety of cell functions via interaction with associated proteins.

## 1. Introduction

The voltage-dependent anion channel 1 (VDAC1), which is located in the outer mitochondrial membrane (OMM), is a multifunctional mitochondrial protein that regulates cell survival and death [1,2,3,4,5]. VDAC1 mediates the transport of adenine nucleotides, calcium, and other metabolites in and out of mitochondria, thereby playing an important role in energy production, cell metabolism, and signaling. It also functions in mitochondria-mediated apoptosis [1,2,3,4,5]. VDAC1 mediates the release of apoptotic proteins located in the mitochondrial intermembrane space via its oligomerization, forming a large channel that allows the passage of cytochrome *c* and apoptosis-inducing factor (AIF) and their release into the cytosol, resulting in apoptotic cell death. VDAC1 also regulates apoptosis via interactions with apoptosis regulatory proteins, such as hexokinase (HK), B-cell lymphoma 2 (Bcl-2), and B-cell lymphoma-extra-large (Bcl-xL), some of which are highly expressed in many cancers [1,4,5].

The location of VDAC1 in the OMM allows it to function in cellular processes via its interaction with proteins involved in metabolism, survival, and cell death pathways [2,6,7]. It is a hub protein that serves as an anchor for about 150 cytosolic, endoplasmic reticulum (ER), nuclear, and mitochondrial proteins, enabling VDAC1 to mediate and regulate the integration of mitochondrial functions with other cellular activities [8,9]. The VDAC1 interactome includes DNA- and RNA-associated proteins, as well as proteins involved in metabolism, apoptosis, signal transduction, and antioxidation [2,7]. VDAC1 interacts with cytoskeletal proteins, such as actin [10,11] and tubulin [12,13], which are involved in mitosis, cell morphology, and focal adhesion. VDAC1, together with its interacting proteins, regulates the integration of mitochondrial functions with other cellular activities, such as metabolism, cell death, and signaling pathways. Thus, VDAC1 appears to be a convergence point for a variety of cell survival and death signals, mediated by its association with ligands and proteins. We developed VDAC1-based peptides that can interfere with these protein–protein interactions, leading to impaired cell metabolism and apoptosis [14,15,16,17,18].

The structure of mammalian VDAC1 was determined at atomic resolution, revealing that VDAC1 is composed of 19 transmembrane β-strands connected by flexible loops to form a β-barrel, with strands β1 and β19 in parallel conformation along with a 26-residue-long N-terminal domain (NTD) that lies inside the pore [2]. However, NTD can be translocated out of the pore [19]. VDAC1 NTD is one of the important structural elements of VDAC1 for which three different structures have been proposed, the variations of which impact its flexibility [20,21,22]: a long helix, small helix, and helix-break helix. The flexibility required for NTD translocation out of the channel’s pore is considered to be provided by a stretch of several glycine residues (^21^GYGFG^25^), connecting the N-terminal domain to β-strand 1 of the barrel [19].

Several different functional roles for the NTD have been proposed including acting as a voltage sensor that regulates channel gating regarding the conductance of ions and metabolites passing through the pore, during VDAC1 dimer formation [19], and possessing an ATP-binding site [23]. Moreover, we showed that expressing the N-terminal truncated form of murine VDAC1 lacking amino acids 1–26 (Δ(1-26)mVDAC1) does not affect cell growth but prevents mitochondrial-mediated apoptosis [3]. The NTD is required for the release of cytochrome *c* and subsequent apoptotic cell death. In addition, (Δ(1-26)mVDAC1 showed no voltage-dependent conductance and exhibited high conductance at all membrane potentials tested [3].

Importantly, NTD mobility is required for VDAC1 dimer formation [19], as well as for the interaction of apoptosis-regulating proteins of the Bcl-2 family (i.e., Bcl-2 associated X protein [Bax], Bcl-2, and Bcl-xL) [1,14,17,19,24], HK-I, and HK-II [3,15].

It should be indicated, however, that VDAC1 in which the N-Ter- α-helix was cross-linked to the wall of the β-barrel pore exhibited typical voltage gating [25]. It is possible that the confirmation of VDAC1-lacking the N-terminus is highly modified, as compared to its presence inside the pore, with no possibility to translocate out of it, a process required for its interaction with associated proteins, such as HK-I, and HK-II, Bcl-2 and Bcl-xL [1,14,15,17,19,24].

We recently engineered VDAC1-based peptides that interfere with the activity of the pro-survival proteins Bcl-2, Bcl-xL, and HK [1,3,5,14,15,16,17]. These VDAC1-based cell penetrating peptides (CPPs) were found to induce cancer cell death in a panel of genetically characterized cancer cell lines, regardless of cancer type or mutation status, with perceived specificity toward cancerous cells [3,15,16]. Studies have demonstrated a triple mode of action: energy and metabolism impairment, interference with the action of anti-apoptotic proteins, and triggering of cell death.

The VDAC1-based peptide, R-Tf-D-LP4, was tested in several cancer types and found to simultaneously attack several hallmarks of cancer, causing impairment of energy, metabolic homeostasis, inhibition of tumor growth, induction of apoptosis, and overexpression of apoptotic proteins [18,26]. Recently, we demonstrated that R-Tf-D-LP4 may be a potential treatment for diabetes mellitus [27], and in addition to liver cancer, it also attenuates non-alcoholic steatohepatitis (NASH) and steatosis [28].

A second peptide, VDAC1-derived N-terminal peptide, was also developed and found to induce cancer cell death [29] and potentially serve as a new therapeutic strategy for amyotrophic lateral sclerosis (ALS) [29].

In this study, we developed several versions of VDAC1-NDT-derived CPPs, focusing on D-Δ(1-18)N-Ter-Antp. We demonstrated that it has multiple effects including apoptosis, autophagy, and senescence induction. In addition, the peptide induces cell division and refusion, resulting in bi-nucleated cells. It also promotes the re-organization of actin and tubulin filaments and alters the expression of signaling proteins.

## 2. Materials and Methods

See the Appendix A for materials. The cell senescence assay, fluorescein isothiocyanate (FITC)-labeled VDAC1-derived D-Δ(1-18)N-Ter-Antp peptide, cell glycogen level [30], and lipid Oil Red O staining were all carried out as described in the Appendix A.

### 2.1. Cell Lines and Culture

PC-3 (human prostate carcinoma), HeLa (human cervix adenocarcinoma), U-87MG (human glioblastoma) were purchased from the American Type Culture Collection (ATCC) (Manassas, VA, USA). MEC-1 (human B-cell chronic lymphocytic leukemia) were from Leibniz institute DSMZ-German collection of microorganisms and cell cultures (Germany). T-Rex-293 (human transformed primary embryonic kidney fibroblasts) were obtained from Thermo Fisher Scientific (Waltham, MA, USA) and MEF (mouse embryonic fibroblast) cell lines were isolated as previously reported [31].

The cells were cultured in an incubator at 37 °C and 5% CO_2_ in recommended growth medium supplemented with 10% fetal bovine serum, 100 U/mL penicillin, and 100 μg/mL streptomycin. Mycoplasma contamination of the cell lines was routinely evaluated.

### 2.2. Peptide Synthesis and Solution Preparation

Customized peptides derived from the VDAC1-N-terminus (Appendix A) were synthesized by GL Biochem (Shanghai, China) to a level of >85% purity. The peptides were dissolved in 100% dimethyl sulfoxide (DMSO) at a concentration of 25 mM, and then diluted with sterile double distilled water to 5 mM (20% final DMSO). Peptide concentrations were determined at an absorbance of 280 nm and a specific molar excitation coefficient, as calculated based on amino acid composition. For all experiments, the final concentration of DMSO in control and peptide-containing samples was ≤0.5%.

### 2.3. Cells Treatment with VDAC1-Based Peptides

MEC-1 cells were incubated in a serum-free medium with various concentrations of VDAC1-based peptides for 1.5 h at room temperature (RT), collected by centrifugation, washed with phosphate-buffered saline (PBS), and analyzed for cell death. Adherent cancer cells were seeded in culture plates and allowed to adhere overnight in a sterile cell culture incubation chamber maintained at 37 °C and 5% CO_2_. The next day, media was removed from the plates, and cells were treated with various concentrations of VDAC1-based peptides for the indicated time in serum-free Dulbecco’s Modified Eagle Medium (DMEM) with and without DMSO (final 0.15%) as a control for 6 h at 37 °C and 5% CO_2_. Cells were trypsinized, centrifuged (1500× *g*, 5 min), and washed with PBS.

### 2.4. Apoptosis Analyses

Cell death was analyzed by the propidium iodide (PI) staining, followed by flow cytometry with the iCyt sy3200 Benchtop Cell Sorter/Analyzer and analysis with EC800 v 1.3.7 software (Sony Biotechnology Inc., San Jose, CA, USA). Apoptosis was analyzed by PI and annexin V-FITC staining, which was carried out according to the manufacturer’s instructions with minor modifications. U-87MG cells, untreated (control) or treated with various concentrations of VDAC1-based peptides, were harvested and washed once with binding buffer (10 mM HEPES/NaOH, pH 7.4, 140 mM NaCl, and 2.5 mM CaCl_2_). Then cells were resuspended in binding buffer containing Annexin V-FITC and incubated for 30 min in the dark. Next, cells were washed and PI (6.25 μg/mL), was added immediately before flow cytometry At least 10,000 events were collected, recorded on a dot plot, and analyzed by flow cytometry.

### 2.5. Immunofluorescence

Cells were seeded on sterile glass coverslips in 12-well plates and cultured until reaching about 80% confluence. Cells were treated with D-Δ(1-18)N-Ter-Antp peptide in serum-free DMEM for 24 h. Then cells were washed with PBS, fixed in 4% paraformaldehyde for 20 min, washed three times with PBS, permeabilized with 0.3% Triton X-100 in PBS, and blocked, in blocking buffer (10% Normal goat serum, 1% fatty acid-free bovine serum albumin [BSA], 0.1% TritonX100 diluted in PBS) for 2 h. Cells were incubated overnight at 4 ℃ with the appropriate primary antibodies (Appendix A), followed by three washes with PBS. Then, cells were incubated with fluorescent-conjugated secondary antibody (Appendix A) for 2 h at RT in the dark. Following a wash with PBS, coverslips were incubated with DAPI (0.5 μg/mL) for 15 min in the dark, and carefully washed, dried, and mounted on slides with fluoroshield mounting medium (Immunobioscience, Mukilteo, Washington, US). After overnight drying at 4 °C, images were acquired using a confocal microscope (1X81; Olympus, Tokyo, Japan).

Quantitation of protein levels, as reflected in the staining intensity, was analyzed in the whole area of the sections using Image J software.

Images were also taken using super-resolution structural Illumination fluorescence microscopy (dual channel N-SIM; Nikon, Vienna, Austria). The SIM-setup used was composed of a 405, 488, 515, 532 and 561 nm excitation laser introduced at the back focal plane inside the SIM-box with a multimodal optical fiber, CFI SR Apochromat TIRF 100× oil (NA 1.49) objective, standard wide field, and SIM filter sets (Nikon, Minato, Tokyo, Japan), and two Andor iXon3 EMCCD cameras mounted to a Two Camera Imaging Adapter (Nikon, Vienna, Austria). At the bottom port, a third CCD camera (CoolSNAP HQ2; Photometrics, Tucson, AZ, USA) was mounted for wide field imaging. For microscope calibration and reconstruction of SIM images, Nikon software (NIS-Element; Nikon, Minato, Tokyo, Japan) was used. Z stacks with 0.12 µm increments were imaged in 4 × 4 to 6 × 6 tiles and stitched together using NIS-elements.

### 2.6. Visualization of Cell Morphology and Division

Cells were seeded in a 6-well plates, treated with D-Δ(1-18)N-Ter-Antp peptide for 24 h, and monitored with the Holomonitor live cell imaging system (Holographic Imaging, Lund, Sweden) for 24 h at 37 °C and 5% CO_2_ or with a microscope (LX2-KSP; Olympus, Shinjuku, Tokyo, Japan), with images captured by a CCD camera.

### 2.7. VDAC1-Based Peptide Interaction with VDAC1-Associated Proteins Using Microscale Thermophoresis

Purified VDAC1 was fluorescently labeled using the NanoTemper Protein-Labeling Blue Kit. Fluorescently labeled VDAC1 was incubated with different concentrations of actin, tubulin, or peptides in 10 mM Tricine buffer (pH 7.4) containing 100 mM NaCl. After a 20-min incubation, 8–10 μL aliquots were loaded into microscale thermophoresis (MST) grade glass capillaries (NanoTemper Technologies, Munich, Germany), and thermophoresis was measured with the NanoTemper Monolith-NT115 system (20/40% light-emitting diode, 40% infrared laser power).

### 2.8. Gel Electrophoresis and Immunoblotting

U-87 MG cells were treated with the indicated concentrations of D-Δ(1-18)N-Ter-Antp peptide in serum-free DMEM for 6 h. Then cells were lysed in lysis buffer (100 mM Tris/HCl pH 8.0, 5% sodium dodecyl sulfate [SDS]), supplemented with a protease inhibitor cocktail (Calbiochem, San Diego, CA, USA). The lysates were vortexed and heated at 70 °C for 10 min. Finally, cell lysates were centrifuged (15,000× *g*, 10 min at 4 °C), and the protein concentration of the supernatant was determined according to the Lowry assay [32] with a slight modification. Protein samples were stored at −80 °C until subsequent gel electrophoresis. Protein samples (20 μg) were resolved by SDS-polyacrylamide gel electrophoresis and immunoblotted using selected primary antibodies (sources and dilutions are listed in Appendix A), followed by incubation with appropriate horseradish peroxidase (HRP)-conjugated secondary antibodies (Appendix A). Proteins were detected using the EZ-ECL Enhanced Chemiluminescence Detection Kit (Biological Industries, Beit Haemek, Israel). Band intensities were analyzed with densitometry using FUSION-FX software (Vilber Lourmat, Collegien, France), and the values were normalized to β-actin.

### 2.9. Actin Purification, Labeling, and Polymerization Assays

Actin was purified from rabbit muscle as CaATP–G-actin complex according to the published method [33] using Supredex S200 (Cytiva Lifesciences) size exclusion chromatography in a buffer G composed of 5 mM Tris/HCl, pH 7.8, 0.2 mM CaCl_2_, 0.2 mM ATP, 0.5 mM dithiothreitol, and 0.01% NaN_3_. The purified G-actin was fluorescently labeled on Cys-374 with N-(1-pyrene) iodoacetamide (Invitrogene) to a molar ratio of 0.8 to 1.0 pyrene/G-actin, as described previously [34]. CaATP–G-actin was converted into MgATP–G-actin by incubation on ice for 20 min with 20 μM MgCl_2_ and 0.2 mM EGTA to 10 μM and was used within 1 h.

G-actin polymerization was monitored by following the increase in the fluorescence of 10% pyrenyl-labeled actin. G-actin polymerization was induced by the addition to CaATP–G-actin solution of a 10-fold concentrated buffer to reach a final concentrations of 50 mM KCl, 1 mM MgCl_2_, 1 mM EGTA, and 10 mM imidazole-HCl, pH 7.0. Fluorescence measurements were carried out at 22 °C in a Horiba/Jobin Yvon FluoroLog Spectrofluorometer using excitation and emission wavelengths of 365 and 407 nm, respectively. Data were analyzed using GraphPad Prism4.1.681 for Windows software (San Diego, CA, USA).

### 2.10. Statistical Analyses

The mean ± SEM obtained from at least three independent experiments are presented. The significance of differences was calculated by a two-tailed Student’s *t*-test and is reported as at *p* < 0.05 (*), *p* < 0.01 (**), *p* < 0.001 (***), or *p* < 0.0001 (****). Data were also analyzed using GraphPad Prism software.

## 3. Results

### 3.1. Human VDAC1 N-Terminal Domain-Based Peptides Induce Cancer Cell Death Regardless of Cancer Type

We previously developed more than 27 different cell-penetrating VDAC1-based peptides as a decoy to compete with VDAC1 for binding to its associated proteins that were tested for apoptosis induction in chronic lymphocytic leukemia (CLL)-derived lymphocytes [16] and in other cell types [26] and evaluated in glioblastoma, breast, and lung cancer mouse models [18,26].

In this study, we produced VDAC1-NTD-derived peptides and developed several versions of cell penetrating NTD-derived peptides (Figure 1A and Appendix A) and tested their apoptosis induction activity. The modifications included shortening of the VDAC1-derived sequences from the C′ or N′ terminus (Appendix A and Figure 1A). We focused on D-Δ(1-18)N-Ter-Antp, composed of the NTD (19–26 amino acids) fused to the CPP Antp, a 16-residue-long sequence from the *Drosophila* antennapedia homeodomain, both containing amino acids in the D-configuration, and examined their cell death induction in several cancer cell lines (Figure 1). Cell death induction, as assayed by PI staining and FACS analysis, was shown for B-cell CLL (MEC-1), PC-3 (human prostate carcinoma), HeLa (human cervix adenocarcinoma), U-87MG (human glioblastoma), and embryonic kidney fibroblast (T-Rex 293) cell lines (Figure 1). The extent of cell death by the various peptides was concentration-dependent and varied between the different cell lines, showing that the non-cancerous T-Rex-293 cell line was the least sensitive (Figure 1D and Appendix A). Thus, the NTD-derived peptides induced cell death in different genetically characterized cancer cell lines regardless of cancer type or mutation status.

The N-Ter Δ(21-26)-Antp, in which the last six amino acids (the GXXXG motif) of the N-terminus were removed, showed no cell death activity (Figure 1B). Thus, this glycine-enriched sequence is required for peptide activity.

To increase N-Ter-Antp peptide stability by reducing its proteolytic degradation, non-native D-form amino acids were used to create the D-N-Ter-Antp peptides. The results showed that D-amino acid peptide was as active as the L-amino acid peptide (Figure 1C–F and Appendix A).

The time course of D-Δ(1-18)N-Ter-Antp peptide cell death induction in U-87MG cells showed that about 70% of cell death was obtained following a 3-h incubation (Figure 1F), compared to chemotherapy drugs, such as cisplatin, which induced cell death after 24–48 h. The FITC-labeled D-Δ(1-18)N-Ter-Antp peptide penetrated the cell, as visualized by confocal microscopy (Figure 1G). The images showed that the peptide entered the cell and induced cell death. Thus, the activity of the NTD-derived peptides was dependent on peptide sequence, concentration, time, and cell type.

For the following studies we selected the D-Δ(1-18)N-Ter-Antp peptide, as it was the most stable, short, apoptosis-inducing peptide, and U-87MG cells, due to their large size that better allowed morphological changes to be monitored.

### 3.2. D-Δ(1-18)N-Ter-Antp Peptide Induces Changes in Cell Morphology and Cytoskeleton Re-Organization

U-87MG cells treated with D-Δ(1-18)N-Ter-Antp peptide showed changes in cell morphology, as reflected by flattening of the cell and enlarged cell areas up to 3-fold relative to untreated cells (Figure 2A,B). The N-Ter Δ(21-26)-Antp peptide showed no cell death activity (Figure 1B), and had no effect on cell morphology (Figure 2C,D).

The peptide-induced changes in cell size were also shown in cells stained for vinculin, a protein that links integrin to the cytoskeleton. Peptide-treated cells showed increased vinculin expression by 4-fold compared to untreated cells (Figure 2E,F). In addition, as a focal adhesion protein, in peptide-treated cells, vinculin was enriched at the cell surface adherent to the matrix, in line with the peptide inhibiting cell migration.

In untreated cells, after 24 h, 64% of the gap was closed, while in peptide-treated cells, depending on its concentration, only 17% of the gap was closed in cells treated with 20 μM of the peptide (Appendix A).

We also detected changes in permeability of HeLa cells upon peptide treatment using the membrane dye FM4-64 (Figure 2G,H). In control cells, as expected, the dye stained the membrane, while cells treated with the peptide showed increased FM4-64 staining, including the cell interior, suggesting that the peptide induced changes in cell membrane permeability. In addition, peptide-treated cells showed small vesicles and membrane blebbing, indicative of apoptosis (Figure 2G, yellow arrows).

### 3.3. D-Δ(1-18)N-Ter-Antp Peptide Induces Apoptosis, Autophagy and Senescence, and Alters the Expression of Apoptosis-, Metabolism-, and Signaling-Related Proteins

The D-Δ(1-18)N-Ter-Antp peptide that induced apoptotic cell death (Figure 1) also increased the expression of proapoptotic proteins, such as caspase-9, AIF, and Bax, while decreasing the expression of anti-apoptotic proteins (Bcl-xL and Bcl-2) (Figure 3A,B).

Next, we tested the possible effects of D-Δ(1-18)N-Ter-Antp peptide on other cell stress-induced effects, such as autophagy and senescence. Autophagy, triggered by stress, such as oxygen, energy or amino acid deprivation, irradiation, or drugs [35,36], was induced by D-Δ(1-18)N-Ter-Antp peptide in U-87MG cells, as demonstrated by the appearance of an autophagy-specific marker, namely the microtubule-associated protein light chain 3 (LC3-II) [37] (Figure 3C,D).

Cellular senescence is induced by diverse types of stress, exhibited by somatic cells losing their capacity to proliferate after a limited number of mitotic divisions and entering cell cycle arrest [38,39]. Cells treated with D-Δ(1-18)N-Ter-Antp showed cellular senescence, which was detected by assessing β-galactosidase activity, a marker of senescence (Figure 3E,F). This was also reflected by increased cell volume [38,39], (Figure 2A–D). Hence, D-Δ(1-18)N-Ter-Antp eliminated cancer cells by inducing autophagy and senescence.

The D-Δ(1-18)N-Ter-Antp peptide also affected cell metabolism by reducing the expression levels of VDAC1, HK-I, ATP synthase 5a, and citrate synthase (CS) (Figure 4A,B). The D-Δ(1-18)N-Ter-Antp peptide also highly enhanced the expression of signaling molecules, such as the putative transcription factor c-Jun (6-fold), nuclear factor kappa B (NF-κB p65) (3.5-fold), and p53 (3-fold; Figure 4C,D), while decreasing the expression of cyclin-dependent kinase 2 (CDK-2) and nuclear factor of the kappa light polypeptide gene enhancer in B-cell inhibitor alpha (IκBα) by 2-fold (Figure 4C,E). Thus, the D-Δ(1-18)N-Ter-Antp altered the expression of various proteins associated with apoptosis, metabolism, cell death, and cell signaling.

To monitor the changes in cell morphology leading to cell enlargement and adherence, actin and tubulin filament organization was detected by actin and tubulin staining with Phalloidin-488 and anti-tubulin antibodies (Appendix A), respectively, followed by confocal and super-resolution microscopy (Figure 5 and Figure 6). U-87MG cells treated with D-Δ(1-18)N-Ter-Antp peptide had more dispersed actin filaments and increased actin levels compared to untreated cells (Figure 5A,B), and the fine actin microfilaments were diminished. Super-resolution microscopy was used to visualize the peptide-induced changes in actin organization with the formation of long fiber-like structure of actin bundles. Different types of actin stress fibers, including dorsal and ventral stress fibers, and transverse arcs were reduced (Figure 5C, I–III). In addition, the cell surface of peptide-treated cells showed many small spikes of actin that might represent filopodia (Figure 5C). Double-nucleated cells were also found in the peptide-treated cells (Figure 5A).

The direct interaction of purified actin with purified VDAC1 was analyzed with microscale thermophoresis (MST) (Figure 5D). Fluorescently labeled VDAC1 was incubated with increasing concentrations of actin, and MST analysis was performed using the NanoTemper Monolith apparatus. By plotting the percentage changes of normalized fluorescence (F Norm %) as a function of compound concentration, a fitted curve yielded a Kd of 30 nM (Figure 5D).

The interaction of the peptide with actin was followed by analyzing its effect on G-actin polymerization (Figure 5E). The fluorescence of polymerized pyrenyl-actin was 7–10 times higher than that of the labeled G-actin monomer [40]. The fluorescence signal was proportional to the actin filament concentration. Therefore, the decrease in the fluorescence signal observed in the presence of the Δ(1-18)N-Ter-Antp peptide indicated an inhibition of the actin filament assembly, and, thus, a direct interaction between the peptide and actin.

As revealed by immunofluorescence staining, peptide-treated cells showed tubulin filament disorganization with the amount of tubulin highly increased (Figure 6A,B). In addition, images from super-resolution imaging revealed that the cells treated with peptide did not have strong microtubule bundles, but rather a fine mesh of fibers (Figure 6C). Direct binding of VDAC1 to purified tubulin analyzed using the MST methods yielded a Kd of 250 nM (Figure 6D). Thus, the peptide induced re-organization of actin and tubulin filaments, and, subsequently, the cell processes associated with actin and tubulin.

### 3.4. D-Δ(1-18)-N-Ter-Antp Peptide Induces Vesicle/Particle Formation

The changes in U-87MG cell morphology in response to the D-Δ(1-18)-N-Ter-Antp peptide also included the appearance of small particles/vesicles in the cells (Figure 7, red circles). Peptide-treated cells that were observed under a light microscope showed a significant number of vesicles/particles (Figure 7A). To identify the nature of these small particles/vesicles present in peptide-treated cells, we considered several options. These small vesicles might represent lysosomes; accordingly, cells were stained with acridine orange (AO), described as trapped in acidic vesicles. In D-Δ(1-18)-N-Ter-Antp peptide-treated cells, an increase of more than 3-fold in orange color was obtained (Figure 7B,C), reflecting AO accumulation in the acidic vesicles [41], most likely representing endosomes and/or lysosomes associated with autophagy (Figure 7B,C).

These small particles/vesicles in peptide-treated cells were visualized by immunostaining with anti-vinculin antibody, showing dark spots all over the cell (Figure 7D,E). The results also showed that in the control, but not in the peptide-treated cells, vinculin was also present in the nucleus (Figure 2E). To better visualize these dark spots, we used super-reso lution microscopy and found that in peptide-untreated cells, vinculin was present in the cell appearing as dots of different sizes that were homogeneously distributed, while vinculin showed different organization with black dots containing no vinculin but was highly enriched around them (Figure 7E). We also considered that the small vesicle-like structures found in peptide-treated cells could be glycogen particles. The results revealed that in D-Δ(1-18)N-Ter-Antp peptide-treated cells, glycogen levels significantly rose with the increase in peptide concentration (Appendix A). Finally, we considered the possibility that the vesicles in peptide-treated cells might represent lipid droplets. Staining with Oil Red O showed that peptide-treated, but not untreated, cells highly accumulated Oil Red O, increasing with incubation time (6, 12, and 24 h) (Figure 7F and Appendix A). In addition, the cells adopted round and large cell aggregates with connections between them, suggesting increased fat production/accumulation or inhibition of its use by the peptide.

### 3.5. D-Δ(1-18)N-Ter-Antp Peptide Alters Cell Division

To monitor the D-Δ(1-18)N-Ter-Antp peptide-induced cell morphological changes, we used a HoloMonitor real-time microscope for single-cell tracking of morphological cell changes induced by the peptide in Hela and U-87MG cells over 48 h, from the early stages to apoptotic cell death. This long-term time-lapse recording demonstrated a very interesting phenomena of cell division with each daughter cell with a nucleus; however, with time, the daughter cells fused (we termed re-fusion), forming a double nuclei cell (Figure 8 and Figure A1).

Monitoring the division–fusion of a single cell showed that the sequence of events started with cell division with nucleated daughter cells, followed by partial nuclei fusion, full nuclei fusion, re-fusion of these cells several hours after cytokinesis, and, finally, membrane blebbing, a hallmark of apoptosis (Figure 8A and Figure A1). Importantly, the cell fusion events occurred almost exclusively between cells sharing the same ancestor (Figure 8A–C and Figure A1) and not between non-related cells. The fused cells increased in size to become giant cells. The percentage of cells with re-fusion of daughter cells was time-, peptide concentration-, and cell type-dependent, with 60–80% of the U-87-MG cells showing this phenomenon following incubation for 24 h with 20 M of the peptide (Figure 8A).

To evaluate whether both daughter cells were completely separated before they re-fused, we monitored the microtubule network during cell division of the microtubule network using specific antibodies against tubulin and phalloidin to label actin (Figure 8B). Both daughter cells remained connected via the midbody until re-fusion, resembling the last connection of dividing cells and consisting of microtubules derived from the central spindle [42] (Figure 8A). Finally, some cells (mainly HeLa) had four nuclei, suggesting that they had residual proliferation potential after fusion (Figure 8C). These effects of the D-Δ(1-18)N-Ter-Antp peptide showed that both daughter cells individually formed an assembled nuclear membrane after mitosis, suggesting complete cell division followed by re-fusion (Figure 8D). Another VDAC1-derived peptide, Tf-D-LP4, showed similar results, inhibiting complete cell division and inducing re-fusion of the daughter cells (Figure A1). The results also showed that the peptide induced HK-I-GFP detachment from the mitochondria.

## 4. Discussion

### 4.1. VDAC1-N-Terminus-Derived Peptide Multiple Effects

The functions of VDAC1, as a mitochondrial governor, are not only due to its transport activity of molecules (up to 5 kDa) but also due to its interaction with over 150 proteins, including cytosolic, ER, and mitochondrial proteins, enabling it to mediate and regulate the integration of mitochondrial functions with other cellular functions [8,9]. Targeting the interactions of VDAC1 with its associated proteins is a promising strategy for the treatment of various diseases. Accordingly, we developed cell-penetrating VDAC1-based peptides as a “decoy” to compete with VDAC1 for its VDAC1-interacting proteins [1,3,14,15,16,17]. These VDAC1-based peptides were derived from one of the VDAC1 cytosol-facing loops, the Tf-D-LP4, and from the VDAC1-N-terminus, both of which induce cancer cell death and inhibit tumor growth [3,15,16,18,27,28,43]. The Tf-D-LP4 peptide has been shown to induce apoptosis in mouse models of glioblastoma [18], and lung and breast cancers [26]. Moreover, the peptide was found to be effective in treating non-alcoholic fatty liver disease [43] and type 2 diabetes [27].

Considering that VDAC1-NTD is required for cell death induction [3], we focused on a selected short version of the VDAC1-N-terminus peptide containing the D configuration amino acid, namely the CPP, D-Δ(1-18)N-Ter-Antp peptide. It interacted with several proteins, including HK-I/II, Bcl-2, Bcl-2-xL [1,3,14,15,16,17], superoxide dismutase 1 (SOD1) [29], and Aβ [44]. It modulated mitochondria-mediated apoptosis while protecting against the death of motor neuron-like NSC-34 cells expressing mutant SOD1 [29]. The peptide also induced multiple effects, including apoptosis, autophagy, and senescence, increased cell adhesion, and altered cell division, possibly via interaction with VDAC1-interacting proteins. The multi-effects of the peptide on cell structure and morphology (Figure 9) might be the result of its interaction with many VDAC1-interacting proteins [8,9]. The VDAC1 N-terminal-derived CPP might have therapeutic potential for cancer and other diseases.

It should be noted that using the BlastP, a sequence homolog of 66% to 88% between the VDAC1-N-terminal-derived sequence FTKGYGFGL and immunoglobulins, glutamate receptors or methyltransferases were found. If this partial sequence homology allows interaction of the peptide with these proteins’ potential partners, then some of the peptide effects might result from interfering with these interactions. In addition, as the VDAC1-N-Ter-derived sequence FTKGYGFGL was also found in VDAC2 and VDAC3, with 7 of the 9 amino acids being identical, some of the presented peptide effects might also result from its competition with VDAC2 and VDAC3 interaction with their partners.

### 4.2. The D-Δ(1-18)N-Ter-Antp Peptide-Induces Apoptosis, Autophagy, Senescence and Alters the Expression of Apoptosis- and Metabolism-Related Proteins

Several versions of the VDAC1-N-terminus-derived CPPs induced apoptosis in various cell lines (Figure 1). The D-Δ(1-18)N-Ter-Antp peptide induced ~90% apoptosis in time- and concentration-dependent manners in U-87MG cells after a 3-h incubation compared to chemotherapy drugs, such as cisplatin, which induced apoptosis only after a 24–48 h incubation. The massive cell death induced by the D-Δ(1-18)N-Ter-Antp peptide might also result from upregulation of pro-apoptotic proteins, such as p53 (Figure 4C,D), caspase-9, AIF, and Bax, decreasing the anti-apoptotic Bcl-xL and Bcl-2 levels (Figure 3A,B), and inducing autophagy and senescence (Figure 3C–F). Senescence, which is triggered in response to a variety of stress conditions [45], might result from peptide effects on metabolism, mitochondrial dysfunction, and apoptosis induction. Cellular senescence inhibition of tumorigenesis is regulated by p53 [46,47], with increased p53 activity and, in some cases, protein levels [48]. We found that p53 was significantly upregulated in peptide-treated cells (Figure 4C,D).

Finally, as previously shown for other VDAC1-based peptides [26,28], the D-Δ(1-18)N-Ter-Antp peptide significantly downregulated the expression of metabolism-related proteins (Figure 4A,B).

### 4.3. D-Δ(1-18)N-Ter-Antp-Induced Morphological Changes Is Associated with the Re-Organization of Actin and Tubulin Filaments

D-Δ(1-18)N-Ter-Antp induced morphological changes, including changes in cell shape, increased cell size, focal adhesion, and re-organization of cytoskeleton actin filaments and microtubules (Figure 5, Figure 6 and Figure 7). Actin filaments and microtubules, both of which interact with VDAC1 (Figure 5 and Figure 6), are important regulators of cell division, migration, and intracellular transport. Mutations or changes in their expression or stability have been linked to cancer progression [49].

The D-Δ(1-18)N-Ter-Antp peptide increased actin and tubulin expression and altered their organization, caused the development of strong filopodia, and reduced the transverse actin stress-fiber arcs, ventral and dorsal stress fibers, and fine actin filaments in the cytosol. In addition, cells treated with the peptide did not have strong microtubule bundles, but rather a fine mesh of fibers. Vinculin staining of cells treated with the peptide showed local spots in which vinculin was absent while it formed a ring structure around these spots. Regardless, the overall staining of vinculin in the peptide-treated cells was increased and a fine network was formed if treated in contrast to isolated clusters in untreated cells.

Using purified VDAC1 and the MST method, we showed, as previously reported, that VDAC1 interacted with actin [10,11] and tubulin [12,13]. Tubulin associates with VDAC1 [13] with dimeric αβ-tubulin inducing closure of bilayer-reconstituted VDAC1 and regulating respiration [12] and permeability to ATP [50]. The VDAC1–tubulin interaction was proposed to serve as a metabolic switch to increase or decrease mitochondrial metabolism, ATP generation, and cytosolic ATP/ADP ratios in [51], and was proposed to sustain the Warburg effect in [52]. G-actin was found to directly and selectively bind to VDAC1 in yeast in [10], reducing conductance of the *Neurospora crassa* VDAC1 channel [11]. No direct interaction of actin with mammalian VDAC1 has been reported. Here, we showed the direct interaction of VDAC1 with purified actin (Figure 5D). The interaction of the peptide with actin was reflected in its inhibitory effect on G-actin polymerization (Figure 5E).

VDAC1 also interacts with other cytoskeletal such as Tctex-1/dynein light chain [53], microtubule-associated protein 2 [54], and gelsolin [55,56]. Gelsolin, which regulates actin filaments, binds VDAC1 in a calcium-dependent manner and inhibits its channel activity and cytochrome *c* release from liposomes [55,56]. The effects of D-Δ(1-18)N-Ter-Antp on cell division, shape, and other activities might result from competing with VDAC1 interaction with cytoskeletal proteins, affecting their polymerization/organization states. Similarly, p53 regulates Rho signaling, which controls actin cytoskeletal organization and prevents filopodia formation, cell spreading, migration, and invasion [57]. Thus, the observed changes in the filament organization [57] might also be mediated by the increased expression of p53 induced by the peptide (Figure 4C,D).

### 4.4. The D-Δ(1-18)N-Ter-Antp Peptide Alters Expression of Signaling-Related Proteins, Modulating IkBα and NF-kB Activities

NF-κB, following activation and translocation to the nucleus [58], coordinates the expression of hundreds of genes in diverse cellular processes, including innate immune response, cell proliferation, and apoptosis [59]. In most cells, NF-κB is complexed with members of the inhibitory IκB family (IkBα, IkBβ, IkBε), which can inhibit its activity by masking its nucleus leading sequence (NLS), thus keeping it in an inactive state in the cytoplasm [60]. Hence, IκΒα exerts proapoptotic activity as it inhibits the anti-apoptotic function of NF-kB. In addition, activated NF-κB (NF-κB p65) targets genes implemented in cell death resistance [61], such as Bcl-xL, A1/Bfl-2, inhibitor of apoptosis proteins (IAP), and tumor necrosis factor receptor-associated factor (TRAF) [62]. In addition, IκB proteins possess an NLS, and can detach NF-κB from the nucleus [63]. The D-Δ(1-18)N-Ter-Antp peptide increased activated NF-κB p65 expression, possibly due to the decrease in IκBα expression (Figure 4C,E). IκΒα interacts with VDAC1 and inhibits apoptosis by stabilizing the VDAC1-HK2 complex and preventing Bax-mediated cytochrome *c* release [64].

The peptide also induced a marked increase in the expression levels of c-Jun (Figure 4C,D), in line with c-Jun expression being induced in response to treatments that frequently trigger apoptosis, such as ultraviolet, ionizing radiation, hydrogen peroxide, and tumor necrosis factor α (TNF-α) [65]. Activated by a large variety of external or internal signals, c-Jun is known to exert effects on cell proliferation, differentiation, cellular transformation, and apoptosis [66]. Overexpression of c-Jun alters cell cycle parameters [67].

Together with Fos family members, c-Jun is a component of the AP-1 transcriptional complex. AP-1 is involved in apoptosis regulation with prolonged and robust c-Jun activation preceding the onset of apoptosis in response to different types of stress signals [68]. The proapoptotic genes induced by c-Jun include TNF-α, FasL, and Bim, the balance and amounts of which determine whether the final outcome results in cell survival or cell death [69]. These facts suggest that peptide-induced apoptosis and upregulating pro-apoptotic proteins might also be involved in overexpression of c-Jun.

### 4.5. D-Δ(1-18)N-Ter-Antp Induces Cell Division and Re-Fusion

Long-term time-lapse microscopy and single-cell tracking of HeLa or U-87MG cell lines demonstrated that, in the presence of the peptide, the divided mononuclear cells re-fused to a single cell with two nuclei, a division-fusion process that was followed by membrane blebbing and apoptosis. Importantly, cell fusion occurred preferentially between cells sharing the same ancestor, and the microtubule network revealed a persistent connection between daughter cells in the majority of re-fusion events (Figure 8 and Figure A1). The finding that primarily daughter cells underwent re-fusion suggested incomplete cytokinesis.

Incomplete cytokinesis and re-fusion of mononucleated daughter cells have been reported [70] and termed acytokinetic mitosis, with nuclear division without complete cellular division causing binocular or multinuclear cells [71]. The mechanism might involve aberrant expression of cell cycle-regulating proteins causing acytokinetic mitosis, but it has not been identified to date. The observation that in peptide-treated dividing cells, daughter cells often remained separated before fusion might argue for a complete separation. However, since cell fusion events occurred almost exclusively between cells sharing the same ancestor, the suggestion was that there was no complete separation. The peptide-induced acytokinetic with many cells showed two nuclei that re-fused, forming a large nucleus and a giant cell that underwent apoptotic cell death, a unique cell death mechanism. The mechanism of neoplastic multinucleation remains unknown, but it is induced by cell–cell fusion or acytokinetic cell division. Acytokinetic mitosis has been reported for re-fusion of small mononuclear Hodgkin cells and is suggested as the mechanism for Reed–Sternberg (RS) cell generation and the formation of giant tumor cells and has been shown in several other lymphoproliferative diseases as well [71,72]. However, in contrast to the development of giant RS, our results showed that the peptide inducing daughter cell re-fusion was followed by apoptotic cell death.

## 5. Conclusions

This study showed the multiactivity of the VDAC1-N-terminal-derived peptide in inducing cell death, autophagy, and senescence, and in altering the expression of proteins associated with cell metabolism, cell signaling, and cell division. Considering that VDAC1 interacts with over 150 proteins involved in many cell functions [8,9], the peptide’s multiple effects can be explained by interfering with the VDAC1 interaction with some of its partner proteins. These results further support the role of VDAC1 as a mitochondrial gatekeeper protein in controlling a variety of cell functions via interaction with associated proteins. Considering that most anticancer drugs target either metabolism, cell signaling cell cycle arrest, apoptosis, or autophagy [73], our findings that D-Δ(1-18)N-Ter-Antp affects all of these processes point to its potential potency as a cancer treatment.

## Figures and Tables

**Figure 1 biomolecules-12-01387-f001:**
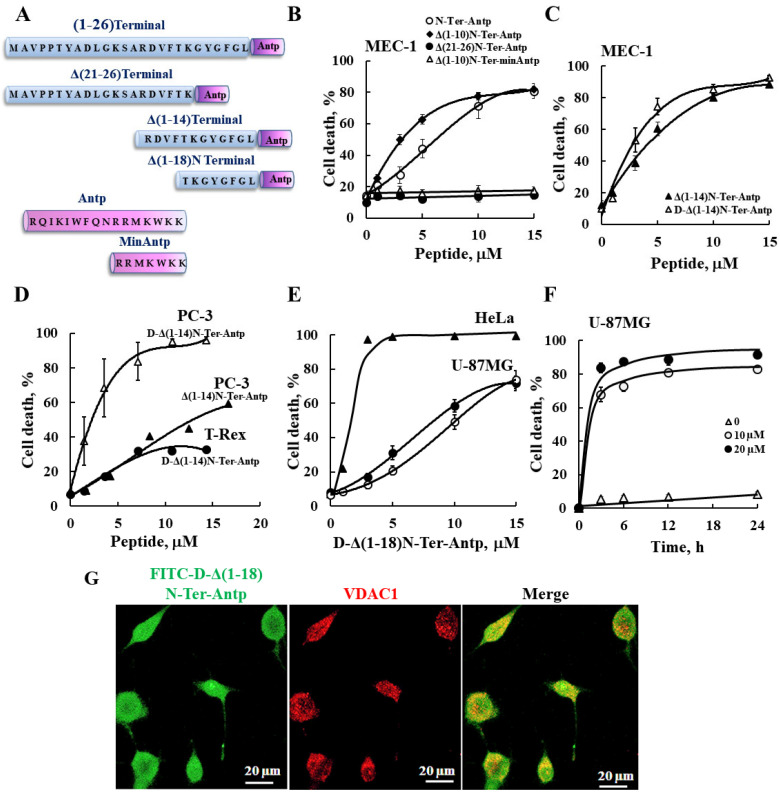
VDAC1-based peptides induced cell death in various cancer cell lines. (**A**) Schematic illustration of the structure of N-Ter-Antp, Δ(1-14)N-Ter-Antp, Δ(21-26)N-Ter-Antp and Δ(1-18)N-Ter-Antp peptides. The VDAC1-derived sequence N-terminus is shown in blue and the CPP sequence Antp is shown in purple. (**B**–**F**) Various cancer cell lines were cultured and incubated with the indicated concentrations of VDAC1-based peptides in the appropriate serum-free growth medium for 6 h at 37 °C, with DMSO at a final concentration of 0.15% in both control and peptide-treated cells followed by analysis of cell death by PI staining and flow cytometry. Data represent mean values ± SE (*n* = 3). (**B**) B-cell CLL (MEC-1) were incubated with the indicated concentrations of N-Ter-Antp (○), Δ(1-10)N-Ter-Antp (♦), N-Ter Δ(21-26)-Antp (●), and Δ(1-10)N-Ter-min-Antp (Δ). (**C**) MEC-1 cells were incubated with the indicated concentrations of Δ(1-14) N-Ter-Antp (▲) and D-Δ(1-14)N-Ter-Antp (Δ) peptides. (**D**) Human prostate cancer cell line (PC-3) (▲, Δ) and human-transformed primary embryonic kidney fibroblast (T-Rex-293) cells (●) were incubated with the indicated concentrations of D-Δ(1-14) N-Ter-Antp peptide (**Δ**, **●**) or with Δ(1-14) N-Ter-Antp peptide (▲). (**E**) Human cervix adenocarcinoma (HeLa) (▲) and human glioblastoma (U-87MG) (●, **○**) cell lines were incubated 6 h with the indicated concentrations of D-Δ(1-18)N-Ter-Antp peptide and analyzed for cell death PI staining and flow cytometry (o,▲) or FITC–annexin V/PI staining and flow cytometry analysis (●). (**F**) U-87MG cells were treated with 10 µM (○), 20 µM (●) or without (Δ) of D-Δ(1-18)N-Ter-Antp peptide for the indicated times, and cell death was analyzed by PI staining and flow cytometry. (**G**) U-87MG cells were incubated for 2 h with FITC-labeled-D-Δ(1-18)N-Ter-Antp peptide (1 mM, green) in serum-free DMEM. Cells were fixed and immuno-stained with anti-VDAC1 antibody (red), and images were captured with a confocal microscope for subcellular localization.

**Figure 2 biomolecules-12-01387-f002:**
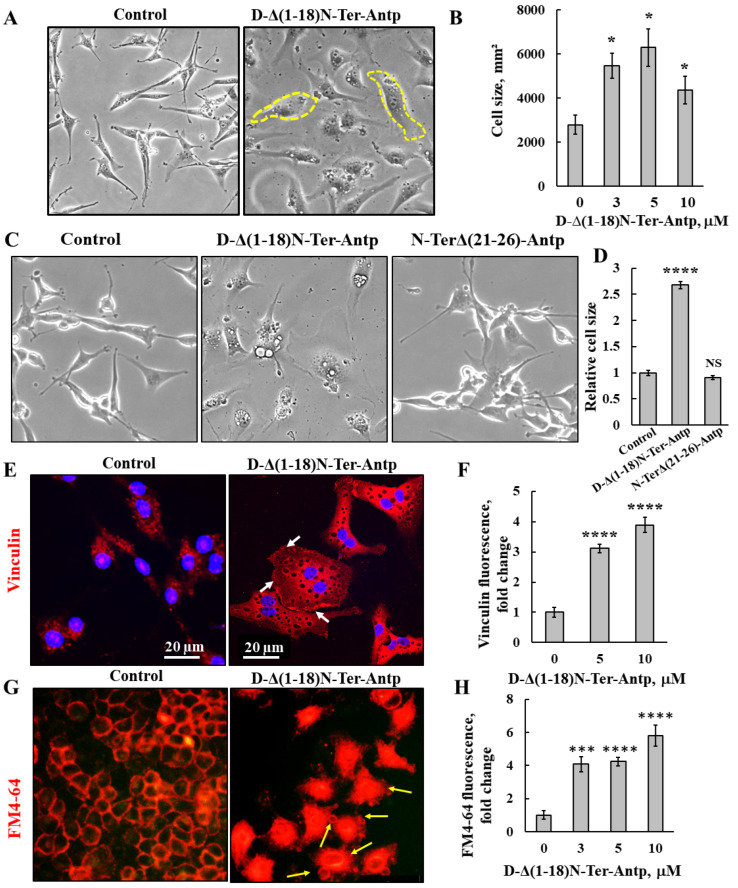
D-Δ(1-18)N-Ter-Antp-induced changes in cell morphology and cell membrane permeability. (**A**,**B**) U-87MG cells were treated with the indicated concentrations of D-Δ(1-18)N-Ter-Antp peptide for 16 h, and cell morphology was monitored by light microscope (**A**). Quantification of cell size (**B**). (**C**,**D**) U-87MG cells were untreated or treated with 10 μM of D-Δ(1-18)N-Ter-Antp peptide or N-TerΔ(21-26)-Antp for 24 h, and cell morphology was monitored by light microscope (**C**). Quantification of cell size (**D**). (**E**,**F**) Cells were stained with anti-vinculin antibodies, followed by secondary anti-mouse-cy3 antibodies. Scale bars are 20 μm. White arrows point to focal adhesion (**E**). Quantification of vinculin staining intensity (**F**). (**G**) D-Δ(1-18)N-Ter-Antp-induced cell membrane permeability changes in HeLa cells. Cells were treated with D-Δ(1-18)N-Ter-Antp peptide (10 µM) for 6 h and subjected to FM4-64 dye staining. Yellow arrows point to apoptotic cells with apoptotic bodies. (**H**) Quantification of fluorescence intensity of FM4-64. * *p* < 0.05, *** *p* < 0.001, **** *p* < 0.0001 (*n* = 3). NS non-specific. The images were captured at 40×, and two folds enlargement is shown.

**Figure 3 biomolecules-12-01387-f003:**
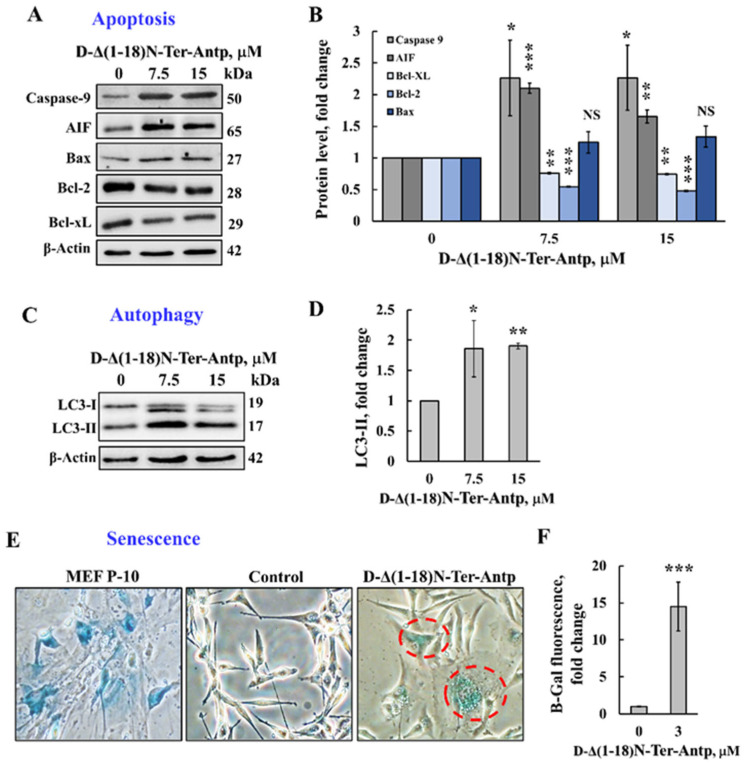
The D-Δ(1-18)N-Ter-Antp peptide induced autophagy and cellular senescence and altered the expression of apoptosis-related proteins. (**A**,**B**) U-87MG cells were treated with the indicated concentrations of D-Δ(1-18)N-Ter-Antp peptide for 12 h and then were subjected to immunoblotting for caspase-9, AIF, Bcl-xL, Bcl-2, and Bax and the relative expression level was quantified (**B**). (**C**,**D**) U-87MG cells were treated with the indicated concentrations of D-Δ(1-18)N-Ter-Antp peptide for 12 h and then subjected to immunoblotting for the autophagy markers LC3-I and LC3-II (**C**) and LC3-II quantification (**D**). (**E**,**F**) U-87MG cells were treated with the indicated concentrations of D-Δ(1-18)N-Ter-Antp peptide for 12 h and then subjected to β-gal activity (blue). Mouse embryonic fibroblast cells at passage 10 (MEF-P10) served as a positive control of senescent cells. Quantification of β-gal staining intensity (**F**). Data are presented as the mean ± SD, *n = 3*, ** p* < 0.05; *** p* < 0.01; *** *p* < 0.001 versus control group. NS, non-specific. The images were captured at 40× and two folds enlargement is shown.

**Figure 4 biomolecules-12-01387-f004:**
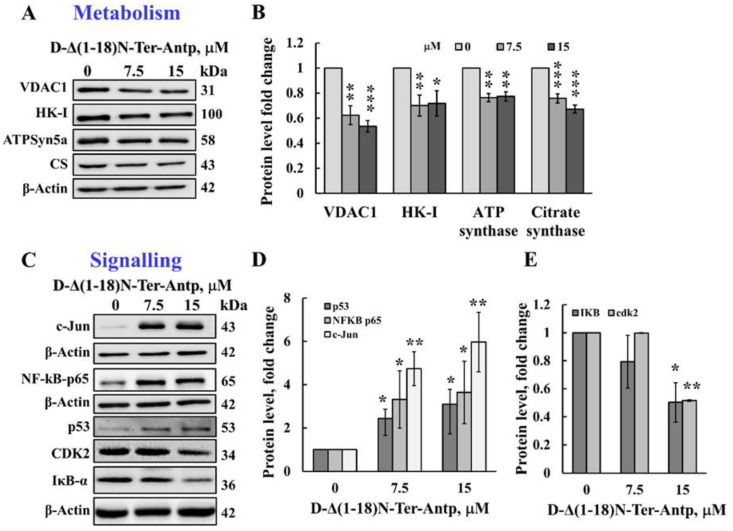
The D-Δ(1-18)N-Ter-Antp induced the altered expression of metabolism and signaling-related proteins. (**A**,**B**) U-87MG cells were treated with the indicated concentrations of D-Δ(1-18)N-Ter-Antp peptide at the indicated concentration for 12 h and then subjected to immunoblot analyses of cell lysates for VDAC1, HK-I, ATPsyn5a, and citrate synthase (CS) (**A**). Quantification of the immunoblot (**B**). (**C**–**E**) Immunoblotting of the indicated proteins c-Jun, NF-κB-p65, p53, CDK2 and IkB-a (**C**) and their quantitative analysis (**D**,**E**). The results were from triplicates of different biological repeats (n = 3) and are presented as the means ± SEM. * *p* ≤ 0.05; ** *p* ≤ 0.01; *** *p* ≤ 0.001.

**Figure 5 biomolecules-12-01387-f005:**
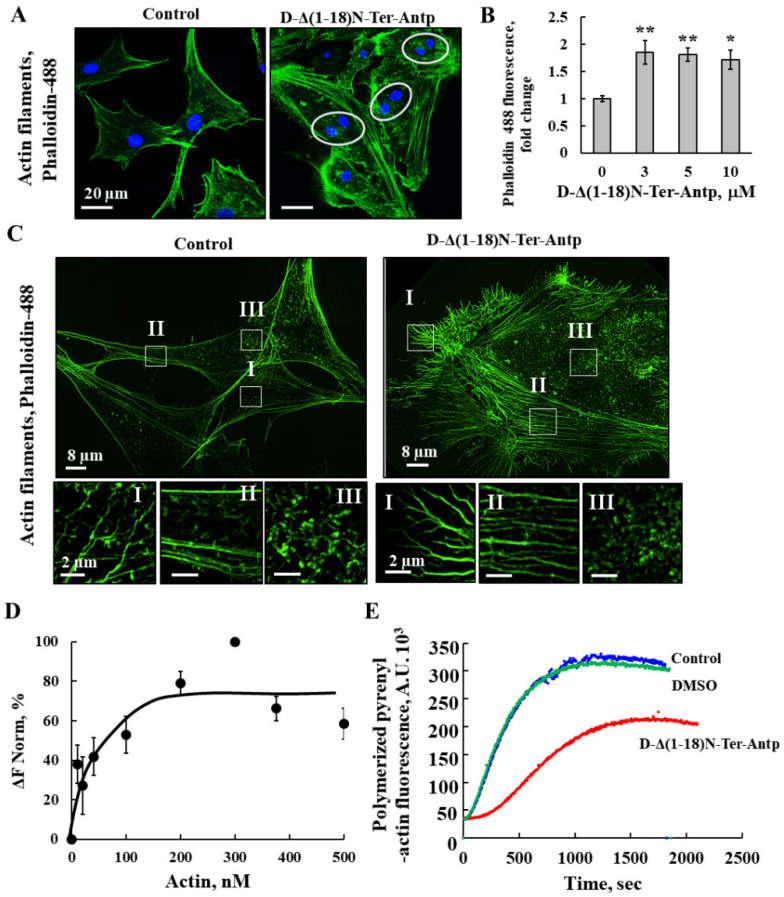
D-Δ(1-18)N-Ter-Antp altered actin filament organization (**A**,**B**) U-87MG cells were cultured and incubated for 24 h with D-Δ(1-18)N-Ter-Antp (10 µM). Cells were stained for actin using Phalloidin-488 and visualized by confocal microscopy (**A**) and quantified for Phalloidin-488 staining intensity (**B**). The results were from triplicates of different biological repeats and are presented as the means ± SEM. * *p* ≤ 0.05; ** *p* ≤ 0.01. (**C**) Maximum intensity projection of super-resolution (SIM) microscopy image. Selected areas (I–III) were enlarged and presented at the bottom. Scale bars were 2 μm. Images show cell sections close to the basal plasma membrane in order to visualize actin structure. (**D**) Actin interaction with VDAC1 analyzed using the MST method. Fluorescently-labeled purified VDAC1 (138 nM) was incubated for 30 min at 37 °C with purified actin (10–500 nM) and then, MST was performed. (**E**) In vitro G-actin polymerization was monitored, as described in the Method section. MgATP–G-actin (2 μM, containing 10% of pyrenyl-labeled actin) was polymerized in the absence (blue line) and presence of 16.7 µM Δ(1-18)N-Ter-Antp peptide (red line) or in the presence of 0.8% DMSO (green line) to a final concentration as in the peptide-containing sample.

**Figure 6 biomolecules-12-01387-f006:**
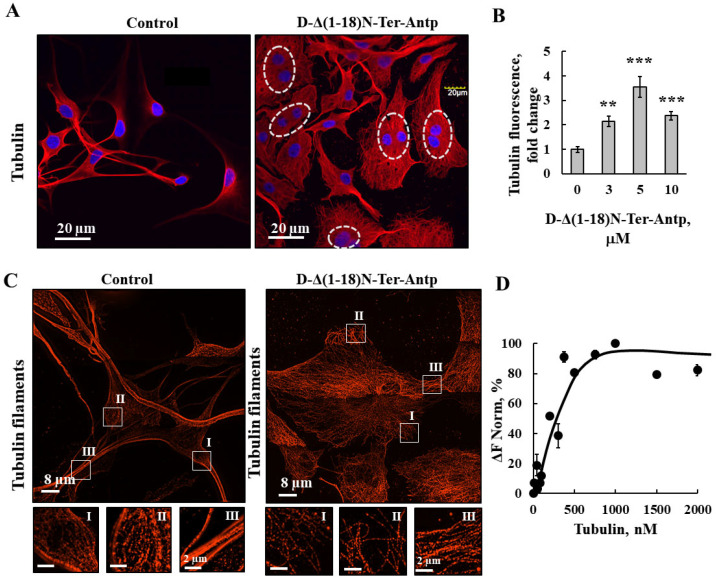
D-Δ(1-18)N-Ter-Antp altered tubulin filament organization. (**A**,**B**) U-87MG cells were cultured and incubated for 24 h with D-Δ(1-18)N-Ter-Antp (10 µM). Cells were subjected to immunofluorescence staining with anti-tubulin antibodies, followed by secondary anti-mouse-Cy3 antibodies, and visualized by confocal microscopy (**A**). Nuclei were stained with DAPI (blue) with circled cells indicate cells with more than one nucleus. Quantification of tubulin staining intensity (**B**). The results were from triplicates of different biological repeats and are presented as the means ± SEM. ** *p* ≤ 0.01; *** *p* ≤ 0.001. (**C**) Maximum intensity projection of super-resolution (SIM) microscopy image. Selected areas (I, II, III) were enlarged and presented at the bottom. Images show cell sections close to the basal plasma membrane in order to visualize tubulin structure. (**D**) Tubulin (20–2000 nM) interaction with VDAC1 analyzed using the MST method.

**Figure 7 biomolecules-12-01387-f007:**
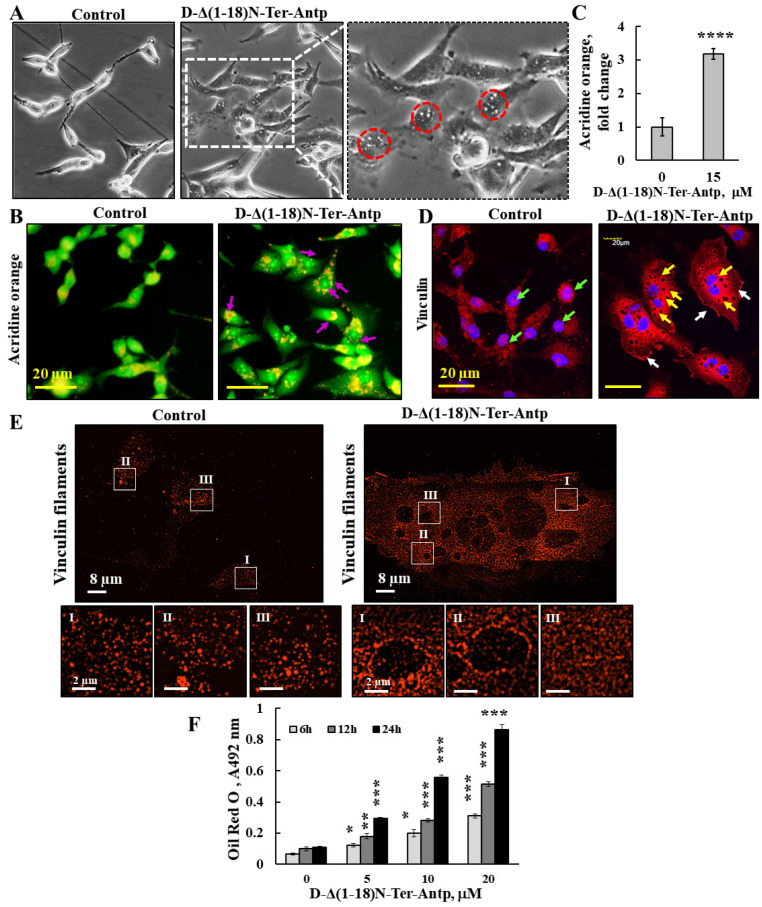
D-Δ(1-18)-N-Ter-Antp peptide induced vesicle/particle formation. U-87MG cells were treated for 6 h with the indicated concentrations of D-Δ(1-18)N-Ter-Antp peptide in serum-free DMEM. (**A**) Cells were visualized for vesicle formation under a light microscope. Red circles indicate vesicles/particles. The images were captured at 40× and two folds enlargement is shown. (**B**,**C**) Cells were stained with acridine orange (AO). Arrows indicate red/orange stained endosomal/lysosomal vesicles (**B**). Scale bars were 20 μm. Vesicles/particles are indicated by violet arrows. Quantification of AO staining intensity (**C**). (**D**) Cells were stained with anti-vinculin antibodies, followed by secondary anti-mouse-cy3 antibodies. Vesicles/particles are indicated by yellow arrows, white arrows point to focal adhesion and green arrows point to vinculin in the nucleus. (**E**) Peptide treated and untreated cells were subjected to IF staining with anti-anti-vinculin antibodies and imaged with SIM super resolution microscopy. Selected areas (I–III) are enlarged and presented at the bottom. (**F**) U-87MG cells were treated with various concentrations of D-Δ(1-18)N-Ter-Antp peptide for the indicated time and subjected to Oil Red O staining and quantification as described in the Appendix A. The results were from triplicates of different biological repeats and are presented as the means ± SEM. * *p* ≤ 0.05; ** *p* ≤ 0.01; *** *p* ≤ 0.001; **** *p* ≤ 0.0001.

**Figure 8 biomolecules-12-01387-f008:**
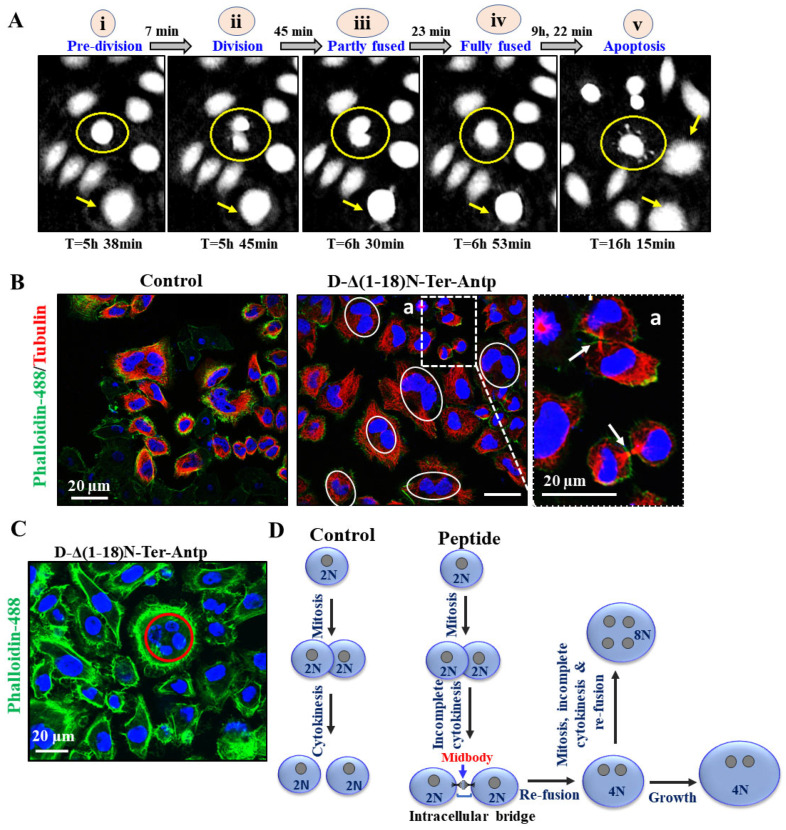
D-Δ(1-18)N-Ter-Antp peptide induced apoptosis and altered cell division. (**A**) HeLa cells were treated with D-Δ1-18N-Ter-Antp (5 μM, 24 h) and monitored using a live-imaging microscope (HoloMonitor). The time course of cell division is presented. Division–fusion phase times are represented in relation to D-Δ(1-18)N-Ter-Antp administration time (T = 0). i. The cells are in a pre-division stage. ii. The cell divides into two cells. iii. The cell becomes partially fused. iv. The cells become fully fused into one large cell. v. Finally, they become an apoptotic cell. (**B**,**C**) HeLa cells incubated with the peptide (10 μM, 24 h) and then stained with anti-tubulin antibodies, followed by secondary anti-mouse-cy3 antibodies (red) followed by actin staining with Phalloidin-488 (green), nuclei stained with DAPI (blue) and visualized by confocal microscopy. Cells with divided nuclei that were connected or in a re-fused state are circled (**C**). (**D**) Schematic illustration of cell division phases of control and D-Δ(1-18)N-Ter-Antp peptide-treated cells.

**Figure 9 biomolecules-12-01387-f009:**
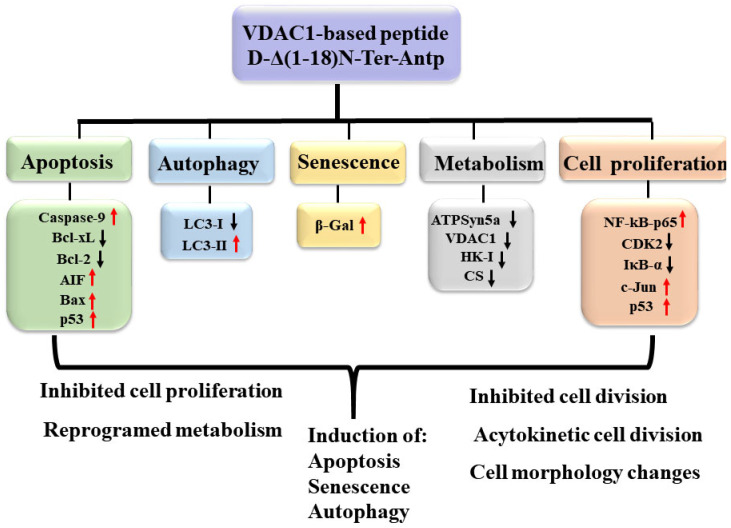
Summary of cell penetrating VDAC1 N-terminal-derived peptide multi-effects. The cell permeable Δ(1-18)N-Ter-Antp peptide targeted multiple signaling pathways involved in apoptosis, autophagy, and senescence induction, and inhibited metabolism and cell proliferation. Red arrows indicate increased expression and black arrows indicate reduced expression.

## Data Availability

Not applicable.

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
