# Peer review of "The Multicellular Effects of VDAC1 N-Terminal-Derived Peptide"

_biomolecules, 2022, doi:10.3390/biom12101387_

Round 1

Reviewer 1 Report

It is well written and documented manuscript indicating the role of targeting the interactions of VDAC1 with cellular proteins as a promising strategy for treatment of some cancers. The targeting could be obtained by using VDAC1 N-terminal-derived peptides. One of the peptide, namely D-Δ(1-18)N-Ter-Antp, was deeply studied within the presented research and obtained results proved its multiple effect on the chosen cancer cell line (U-87MG).

Comments for the authors:

1.  VDAC is located in OM, not at OM (line 39)

2. Figure 1 could be prepared in a more careful way; accordingly it is not clear why the peptide and the cell line U-87MG were chosen for further research.

3. It is also not clear why MEC-1 cell line is mentioned in the Materials and Methods section and why in some experiments U-87MG cells are replaced by HeLa cells.

4. Overall, the Material and Methods section could be rewritten to adequally address presented experiments

5. The issue of different sensitivity of different cancer cell lines to different VDAC1 N-terminal-derived peptides could be discussed.

4. The figure is not specified (line 260)

Author Response

Comments for the authors:

Comments for the authors:

  1.  VDAC is located in OM, not at OM (line 39), corrected
  2. Figure 1 could be prepared in a more careful way; accordingly it is not clear why the peptide and the cell line U-87MG were chosen for further research.

This figure attempted to present the activity of several VDAC1-N-terminal-derived peptides in cell death induction that are not cell-type dependent. As indicated, we previously used VDAC1-N-terminal-derived peptides. Here, we further designed additional peptides, aiming to identify the most stable, short, apoptosis-inducing ones. Accordingly, the D-Δ(1-18)N-Ter-Antp peptide fit these requirements and was selected for further study.

As for selecting U-87MG, although the peptide induced apoptosis in different genetically characterized cancer cell lines regardless of cancer type or mutation status, we selected U-87MG because it is the larger cell, and the morphological studies are well observed.

We now added the following to the MS (lines 300-302):

For the following studies we selected the D-Δ(1-18)N-Ter-Antp peptide as it is the most stable, short, apoptosis-inducing peptide and U-87MG cells due to their large size that better allows  morphological changes to be monitored.

  1. It is also not clear why MEC-1 cell line is mentioned in the Materials and Methods section and why in some experiments U-87MG cells are replaced by HeLa cells.

This cell line was used in this study (Fig. 1B, C) to represent cells in suspension, while the other cell lines used are adherent.

  1. Overall, the Material and Methods section could be rewritten to adequally address presented experiments

 We have modified this section.

  1. The issue of different sensitivity of different cancer cell lines to different VDAC1 N-terminal-derived peptides could be discussed.

In general, as summarizing in Table S3, the sensitivity of the different cancer cell lines to the active peptides was similar with IC50 between 2 and 5 µM.

Reviewer 2 Report

In the present manuscript Anand et al. explore the effects of a cell penetrating VDAC-derived peptide on various cellular functions. Although potentially interesting I see major weaknesses mainly due to the lack of appropriate controls.

I have two major and a few minor comments.

Major remarks

My most important point is the lack of appropriate controls. This includes:

a) negative controls for experiments in Figs 2-8. Although the authors present a few ineffective peptides in Figure 1, they use untreated cells as a control for all experiments in Figs 2 through 8. Considering that most of the observed effects like changes in morphology, blebbing or the formation of intracellular vesicles are fairly general signs of unhealthy cells, control cells with an inert peptide like e.g. a 9 alanine peptide linked to Antp should be used to exclude that the observed effects are simply due to the presence of exogenous peptides, rather then effects related to the lack of interaction of VDACs with partner proteins.

b) a control for the specificity of the peptide. The authors use a fairly short peptide (9aa) from the N-Terminus of VDAC. A simple blastp search using the sequence FTKGYGFGL retrieves several proteins with high homology including immunoglubulins, glutamate receptors or methyltransferases. It is thus not clear, weather the observed effects are indeed attributable to the lack of VDAC with interaction partners. Although VDAC1-null cells were described to be apoptotic by themselves it would still be interesting to see if treatment of these cells with the peptide can induce further effects. This should not happen according to the authors hypothesis.

c) a control of the efficacy of the peptide. The authors present MST data to present a direct interaction of VDAC1 with tubulin and actin. According to the hypothesis of the authors, namely that the presence of the peptide blocks interaction with partner proteins, these experiments should be repeated in the presence of the peptide to show a lack of binding.

My second major point is that the authors attribute all of their results specifically to VDAC1. There are three isoforms of VDAC which are expressed in almost all cells (expression analysis of the used cell lines by qPCR could be helpful to further characterize them). The peptide sequence FTKGYGFGL is conserved among isoforms and species and the antibody used in figures 1 and 3 (ab15895) detects all three isoforms according to the manufacturer. There is thus no specificity for VDAC1 in the experiments. All observed effects (considering they are specific to VDAC - see my previous point b) have to be interpreted to be a result of the lack of interactions by VDACs not VDAC1. In contrary, it is a charming hypothesis that individual effects are attributable to individual isoforms, which could be subject of future studies.

Minor remarks

line 235: The authors write: “The N-Ter Δ(21-26)-Antp in which the last four amino acids (the GGGG motif) of the N-terminus were removed...”. However, six amino acids including only three glycines were removed (GYGFGL). This should be corrected.

line 83-84. The authors cite their own work in which removal of the NTD abolished gating. In this context previous work indicating that affixing the helix to the channel wall does not interfere with gating (Teijido et al, JBC, 2012) should be critically discussed.

In all figures significance stars should be centered above the bars

Author Response

Major remarks

My most important point is the lack of appropriate controls. This includes:

  1. negative controls for experiments in Figs 2-8. Although the authors present a few ineffective peptides in Figure 1, they use untreated cells as a control for all experiments in Figs 2 through 8. Considering that most of the observed effects like changes in morphology, blebbing or the formation of intracellular vesicles are fairly general signs of unhealthy cells, control cells with an inert peptide like e.g. a 9 alanine peptide linked to Antp should be used to exclude that the observed effects are simply due to the presence of exogenous peptides, rather then effects related to the lack of interaction of VDACs with partner proteins.

We believe that the best negative control is the N-terminal peptide lacking the GXXXG motif (D(21-26)-N-Ter-Antp), showing no pro-apoptotic activity. This peptide had no effect on cell morphology as reflected in no change in cell size and no appearance of intracellular vesicles, now shown in new figure Fig. S1. This suggests that that the cellular effects induced by the other peptides are not due “to the presence of exogenous peptides”.

To present these results earlier in the paper, we moved Fig. 4, presenting the effect of the peptide on cell morphology, to be Fig 2.

  1. a control for the specificity of the peptide. The authors use a fairly short peptide (9aa) from the N-Terminus of VDAC. A simple blastp search using the sequence FTKGYGFGL retrieves several proteins with high homology including immunoglubulins, glutamate receptors or methyltransferases. It is thus not clear, weather the observed effects are indeed attributable to the lack of VDAC with interaction partners. Although VDAC1-null cells were described to be apoptotic by themselves it would still be interesting to see if treatment of these cells with the peptide can induce further effects. This should not happen according to the authors hypothesis.

It is very interesting that this reviewer found that the VDAC1-derived sequence FTKGYGFGL is present with high homology in other proteins. Our BlastP search (see below) identified only VDAC1, some hypothetical proteins, and beta-lactamase family proteins, which are bacterial proteins.

 For best interaction/competition, an identical sequence is required to interfere with the interaction with a protein, and then the D-Δ(1-18)N-Ter-Antp peptide may weakly  interfere with the interaction of these proteins with their potential partners. However, in this short peptide, changes in a single amino acid may alter the interaction of the peptide with these potential partners. We and others have found, for example, that a single mutation in VDAC1 E72Q prevented hexokinase binding to VDAC1 (Ref 1 in the MS).

We have added the following to the Discussion:

Using the BlastP, a sequence homolog of 66% to 88% between the VDAC1-N-terminal-derived sequence FTKGYGFGL and immunoglobulins, glutamate receptors or methyltransferases were found. If this partial sequence homology allows interaction of the peptide with these proteins’ potential partners, then some of the peptide effects may result from interfering with these interactions.

As this reviewer indicated, the peptide competes with VDAC1 interaction with its partners and does not affect VDAC1 levels. To our knowledge, VDAC1 is required for apoptosis as VDAC1 silencing by siRNA efficiently prevented cisplatin-induced apoptosis and Bax activation in non-small cell lung cancer cells (Tajeddine, N.et al and Kroemer, G. (2008) Hierarchical involvement of Bak, VDAC1 and Bax in cisplatin-induced cell death, Oncogene 27, 4221-4432). These results indicate that VDAC1 is an essential protein for apoptotic cell death, and its absence inhibited rather than induced apoptosis.

  1. c) a control of the efficacy of the peptide. The authors present MST data to present a direct interaction of VDAC1 with tubulin and actin. According to the hypothesis of the authors, namely that the presence of the peptide blocks interaction with partner proteins, these experiments should be repeated in the presence of the peptide to show a lack of binding.

The proposed experiment is simple to do, but very complicated to analyze, as different complexes are formed such as peptide with actin, and actin with VDAC1.  

We, however, added a new experiment in which the effect of the peptide on G-actin polymerization was analyzed (Fig. 5E). The presence of the Δ(1-18)N-Ter-Antp peptide inhibited actin filament assembly, suggesting a direct interaction between the peptide and actin.

My second major point is that the authors attribute all of their results specifically to VDAC1. There are three isoforms of VDAC which are expressed in almost all cells (expression analysis of the used cell lines by qPCR could be helpful to further characterize them). The peptide sequence FTKGYGFGL is conserved among isoforms and species and the antibody used in figures 1 and 3 (ab15895) detects all three isoforms according to the manufacturer. There is thus no specificity for VDAC1 in the experiments. All observed effects (considering they are specific to VDAC - see my previous point b) have to be interpreted to be a result of the lack of interactions by VDACs not VDAC1. In contrary, it is a charming hypothesis that individual effects are attributable to individual isoforms, which could be subject of future studies.

The VDAC1-N-Ter-derived peptide sequence shows a high, but not complete, homology (7 of the 9 amino acids) with a sequence in the N-terminal of VDAC2 and VDAC3.

VDAC1   FTKGYGFGL

VDAC2  FNKGFGFGL

VDAC3  FNKGYGFGM

Indeed, it is possible that some of the Δ(1-18)N-Ter-Antp effects also involve peptide competition with VDAC2 and VDAC3 interaction with their partners, although for protein–protein interaction, the differences in the sequences may be significant.

We have shown that the peptide interacts with actin and prevents polymerization involving protein–protein interactions (Fig. 5E).

As proposed by this reviewer, this is a subject for future studies, and it requires identifying VDAC2 and VDAC3 interactomes.

We have added the following to the Discussion:  As the VDAC1-N-Ter-derived sequence FTKGYGFGL is found also in VDAC2 and VDAC3, with 7 of the 9 amino acids identical, some of the presented peptide effects may also result from its competition with VDAC2 and VDAC3 interaction with their partners (lines 562-565).

Minor remarks

line 235: The authors write: “The N-Ter Δ(21-26)-Antp in which the last four amino acids (the GGGG motif) of the N-terminus were removed...”. However, six amino acids including only three glycines were removed (GYGFGL). This should be corrected.

Sorry about indicating GGGG as the motif and not GXXXG (New findings concerning vertebrate porin II — On the relevance of glycine motifs of type-1 VDAC Molecular Genetics and Metabolism 108 (2013) 212–224).  This is corrected now

lines 83-84. The authors cite their own work in which removal of the NTD abolished gating. In this

We have added the following:

It should be indicated, however, that VDAC1 in which the N-Ter-α-helix was cross-linked to the wall of the β-barrel pore exhibited typical voltage gating (Teijido et al. BIOENERGETICS|  287, ISSUE 14, P11437-11445). It is possible that the confirmation of VDAC1 lacking the N-terminus is highly modified, as compared to its presence inside the pore, with no possibility to translocate out of it—a process required for its interaction with associated proteins such as HK-I and HK-II, Bcl2, and Bcl-xL (lines 89-93).

In all figures significance stars should be centered above the bars

Thanks, done

Round 2

Reviewer 2 Report

The authors have addressed most of my concerns, however some doubts remain.

The authors now show effect on cell morphology as figure 2 which greatly enhances the logical flow of the manuscript. They also show an appropriate control in Figure S1, however in my opinion the control should be shown in a main figure (Fig2) and with the corresponding quantitative analysis (as done in figure 2B).

The authors have included a new experiment showing that the peptide inhibits the polymerization of actin. This proofs, as correctly stated by the authors, “a direct interaction between the peptide and actin”. However, it does not indicate that the peptide indeed reduces interaction of partner proteins with VDAC, an experiment that is still missing. Since the authors use labeled VDAC in their MST experiments, the peptide-actin complex is non-fluorescent and should not be visible in the experiment. Alternatively, FRET approaches or high-resolution microscopy to show the lack of co-localization could be envisioned.

The authors have added a sentence about possible effects being induced by the lack of VDAC2 and VDAC3 interactions to the discussion. This sentence should be presented in the introduction or at the beginning of the results, so that the reader can keep this caveat in mind during the reading process.

Figures 1 and 4: Antibody ab15895 is a panVDAC antibody, signals in ICC and WB should be correctly labeled as VDAC.

Line 266. The last six amino acids were removed, not four.

Author Response

The authors now show effect on cell morphology as figure 2 which greatly enhances the logical flow of the manuscript. They also show an appropriate control in Figure S1, however in my opinion the control should be shown in a main figure (Fig2) and with the corresponding quantitative analysis (as done in figure 2B).

As suggested, the results in Fig. S1 and their quantification were now presented as Fig. 2C,D

The authors have included a new experiment showing that the peptide inhibits the polymerization of actin. This proofs, as correctly stated by the authors, “a direct interaction between the peptide and actin”. However, it does not indicate that the peptide indeed reduces interaction of partner proteins with VDAC, an experiment that is still missing. Since the authors use labeled VDAC in their MST experiments, the peptide-actin complex is non-fluorescent and should not be visible in the experiment.

 We have carried out the suggested experiment and found that the peptide binds not only to actin but also to VDAC1. This is possible considering our previous study showing that the N-terminal domains from 2 molecules in the dimer/oligomer can interact with each other (Geula S;  Ben-Hail, D;  Shoshan-Barmatz, Structure-based analysis of VDAC1: N-terminus location, translocation, channel gating and association with anti-apoptotic proteins V. Biochem J (2012) 444 (3): 475–485). This is shown using cysteine-lacking VDAC1, cysteine residue substitution (as in the N-terminal domain), and using a thiol-specific cross-linker, showing that VDAC1 can dimerizes with the N-terminal domain of a second VDAC1 molecule.

Alternatively, FRET approaches or high-resolution microscopy to show the lack of co-localization could be envisioned.

This is not simple experiment to do as to visualize the peptide it must be labeled and addition of any probe to the very small peptide would change its interaction with VDAC1 partners.

The authors have added a sentence about possible effects being induced by the lack of VDAC2 and VDAC3 interactions to the discussion. This sentence should be presented in the introduction or at the beginning of the results, so that the reader can keep this caveat in mind during the reading process.

We appreciate this reviewer above suggestion, but we believe that indicating this possibility in the discussion is the best place, as it is offered in content of presenting the similarities in the N-terminal sequences of the 3 isoforms, indicating that it is not 100% homology.

Figures 1 and 4: Antibody ab15895 is a panVDAC antibody, signals in ICC and WB should be correctly labeled as VDAC.

Done

Line 266. The last six amino acids were removed, not four.

 Done